# Accuracy Verification of Surface Models of Architectural Objects from the iPad LiDAR in the Context of Photogrammetry Methods

**DOI:** 10.3390/s22218504

**Published:** 2022-11-04

**Authors:** Piotr Łabędź, Krzysztof Skabek, Paweł Ozimek, Dominika Rola, Agnieszka Ozimek, Ksenia Ostrowska

**Affiliations:** 1Faculty of Computer Science and Telecommunications, Cracow University of Technology, Warszawska 24, 31-155 Kraków, Poland; 2Faculty of Architecture, Cracow University of Technology, Warszawska 24, 31-155 Kraków, Poland; 3Faculty of Mechanical Engineering, Cracow University of Technology, al. Jana Pawła II 37, 31-864 Kraków, Poland

**Keywords:** photogrammetry, iPad, point cloud, 3D reconstruction, architectural objects, quality verification, LiDAR

## Abstract

The creation of accurate three-dimensional models has been radically simplified in recent years by developing photogrammetric methods. However, the photogrammetric procedure requires complex data processing and does not provide an immediate 3D model, so its use during field (in situ) surveys is infeasible. This paper presents the mapping of fragments of built structures at different scales (finest detail, garden sculpture, architectural interior, building facade) by using a LiDAR sensor from the Apple iPad Pro mobile device. The resulting iPad LiDAR and photogrammetric models were compared with reference models derived from laser scanning and point measurements. For small objects with complex geometries acquired by iPad LiDAR, up to 50% of points were unaligned with the reference models, which is much more than for photogrammetric models. This was primarily due to much less frequent sampling and, consequently, a sparser grid. This simplification of object surfaces is highly beneficial in the case of walls and building facades as it smooths out their surfaces. The application potential of the IPad LiDAR Pro is severely constrained by its range cap being 5 m, which greatly limits the size of objects that can be recorded, and excludes most buildings.

## 1. Introduction

Methods of restoring three-dimensional models have been widely used in various fields of research. Three-dimensional modelling is becoming more and more popular in the design and surveying of architecture and landscape-architecture objects, replacing traditionally used two-dimensional drawings and photographs [1]. The precision of the resultant models, in this case, translates into reliable insight about objects. This information can be used in popularising works of art, and can play an educational role [2,3], as well as form a basis for conservation and reconstruction works [4]. In such cases, it supports an entire range of activities, covering the following phases: construction work planning, creating technical documentation and cost estimation, up to construction work supervision and compliance with design documentation. A precisely developed model can be a key element of building information management (BIM) [5], and in the case of historic buildings, historical building information management (HBIM) [6,7]. It also allows the monitoring of the technical condition of a historical site and possible changes in its structure occurring over time [6,8].

Emerging sophisticated tools allow either partial or complete automation of the modelling process, so it is possible to obtain a 3D reconstructions based on satellite data [9], stereoscopic images [10], or light detection and ranging (LiDAR) data [11,12,13,14]. Photogrammetric reconstruction is also widely used (e.g., [2,15,16,17,18]). It consists of the restoration of the position and the spatial relations between 3D points of observed surfaces on the basis of 2D images [19]. Such an approach is often referred to as structure from motion (SfM) [20].

Photogrammetry has become generally available thanks to developing applications that use optical sensors installed in mobile devices [21]. This applies to both photographs obtained at human eye level and those collected by unmanned aerial vehicles (UAVs) (e.g., [16,22,23,24,25,26,27,28]). Due to their relatively easy availability, models created by photogrammetric methods are used in a broad spectrum of different fields, ranging from the digitisation of museum collections [29], through feasibility studies and construction planning [30], to applications in the entertainment industry [31]. Due to such a wide dissemination of the method, questions arise about the accuracy of the models created in this way [16,28,32,33], research on their verification by other methods (e.g., [16,24,34,35]), or attempts to increase their precision [36,37].

LiDAR is a technology that uses beams of electromagnetic radiation to generate information about objects. It has been associated mainly with large-scale applications, such as forestry, mining, oceanography, archaeology, topography, land surveying, and urban planning [38,39]. The development of consumer devices such as smartphones and tablets has shown that this technology can also be applied in mobile devices. LiDAR, in this case, is based on time of flight (ToF) technology, which determines the time it takes for a pulse or modulated light signal to travel a distance from an object [40,41]. Currently, there are several mobile devices that use these technologies (LiDAR, ToF, optical sensors) on the market [42]. This paper discusses research conducted by using an Apple iPad Pro device (Apple Inc., Cupertino, CA, USA) [43], whose LiDAR sensors are based on direct time of flight (dToF) technology [44].

To the best of our knowledge, there have been no comprehensive studies in the literature concerning the verification of the quality of 3D reconstructions derived from LiDAR sensors in iPad devices, especially in the context of architectural and landscape objects. A comparative analysis of the accuracy of scans from this device with scans using an industrial 3D scanner has been presented in [40]. However, it applied to objects of minuscule size. A slightly more extensive study was presented by Gollob et al., whose use of an iPad device was mentioned in investigating the accuracy of forest surveying variables [45]. Heinrichs and Yang presented an analysis in which they verified the bias and repeatability of the 3D scans made by using the device [46]. One can also find various types of comparative analyses online in blogs or vlogs, but they do not have the value of peer-reviewed academic studies. Therefore, it is expedient to demonstrate the iPad LiDAR’s potential to create surface models and to indicate types of built objects that could be digitised in this way, as no similar studies were found in the literature.

The purpose of this paper is to compare 3D scans acquired by using the Apple iPad Pro LiDAR sensor with photogrammetric reconstructions. Measurement accuracy was compared. Analyses were performed for models of four types of architectural elements: (1) a small detail of significant geometric complexity, (2) a piece of garden furniture (garden sculpture), (3) an architectural interior, and (4) a building’s facade. The measurement capabilities of the devices under study in relation to the groups listed are presented, and the quality of the obtained measurements is summarised. This study allowed for formulating recommendations on the use of the Apple iPad Pro in various applications. The theoretical contribution of this study was to propose and validate a method for comparing surface models based on positional statistics.

## 2. Materials and Methods

### 2.1. Accuracy of Reference Object Measurements

A set of calibration balls from the Laboratory of Coordinate Metrology (at the Cracow University of Technology) was used to measure the accuracy of the scanning device (Figure 1).

The set consisted of two polished and tarnished metal balls $d_{1}$ and $d_{2}$. Their diameters were $d_{1}=85.02$ mm and $d_{2}=85.01$ mm respectively, with a shape deviation of $\delta_{k}=0.02$ mm. The spheres were placed on a metal frame at a distance of $d_{k}=269.4$ mm. The listed values have been confirmed on a calibration certificate by an accredited laboratory. The measurement set allowed for the precise verification of the quality of the spatial representation of the objects. A sphere size measurement strategy was adopted that consisted of:(1)manually marking parts of the mesh that represented the spheres,(2)determining the equation of the sphere fitted to the marked area by using the root mean square (RMS) error minimisation algorithm. In this way, two spheres were created in the observation area, for which their centre distances could be determined, and(3)after a series of measurements, determining the average value of the distance between the observed spheres $d_{av}$; this quantity was compared against the nominal value of $d_{k}$.

If the measurement error reaches a small value, this indicates the potential suitability of the device for generating accurate 3D models of built objects.

### 2.2. Input Data

In order to carry out a precise analysis of the actual capabilities of the device as mentioned above in the context of 3D scanning, an approach was adopted that involved the selection of objects with different spatial scales. Such a study allows one to determine the accuracy that the device can achieve for different applications, but also to identify applications whereby the use of the device can give correct results. Indeed, although the maximum range of the LiDAR sensor is specified, there is no mention of its resolution. It is therefore unclear whether fine details are imaged correctly and with sufficient precision.

With this in mind, analyses were conducted for four sites, each representing a different scale and type of architectural work.
1.**Fine detail.** The item analysed was a plaster figurine covered with paint, characterised by fine detail and surface irregularities. Its overall dimensions are 99.2 × 40.8 × 90 mm. This type of form may represent the finest architectural detail found in ornamental elements, such as window and door frames or stucco mouldings. The small size meant that, together with the actual object, a large area of the surroundings was also collected. This, in turn, entailed the necessity of appropriate selection of the scanning location (Figure 2a).2.**Small architectural object.** The bust of Tadeusz Kosciuszko—a Polish national hero and patron of the Cracow University of Technology—placed on a granite pedestal, was chosen as the second test object. The bronze monument, designed by Professor Stefan Dousa, is located in the central part of the university’s campus, surrounded by trees and footpaths. This brings greenery elements within the theoretical range of the LiDAR sensor. Both the location and the material from which the sculpture is made can present challenges for photogrammetric methods due to the potential for significant differences in the illumination of the different parts. The dimensions of the statue are 678 × 458 × 754 mm (Figure 2b).3.**Indoor space.** The interior shown in (Figure 2c) is a fragment of a hall leading to lecture rooms, located in a modernist building of the Faculty of Chemical Engineering and Technology at the Cracow University of Technology (design: E. Moj, building in use since 1970). It was chosen as a test object for several reasons. Its area is so large that it is impossible to cover it all only from one point, so it is necessary to move with the device. In addition, many elements in the corridor may be difficult to represent on a 3D model, such as structural columns, as well as one of the walls, the ceiling beams, fixtures, and glazed doors. The dimensions of the room are 690 × 525 × 275 cm.4.**Building facade.** One of the Cracow University of Technology’s campus buildings was used to analyse the LiDAR sensor’s performance in outdoor scenes with an object of significant size. Purchased by the university in 2010, this 1918 military building was initially used as an artillery equipment warehouse and later as a military bathhouse and laundry room. It has been adapted for teaching purposes while retaining its original external form. The main problem, in this case, were the dimensions, which exceed the theoretical range of the sensor. For a slight simplification, only one of the elevations (the southern one, which was preserved as a “witness to history” during the building’s adaptation while the others were remodelled) was selected for comparison. However, it posed a challenge to the methods that were tested because of the historical architectural detailing that gives it a spatial, three-dimensional form. In the articulation of the peak elevation of the two-story building, covered with a gable roof, four symmetrically spaced pilasters stand out, and divide it into three fields. The central part contains two windows (one above the other) with arched brick lintels. Above them, there is a small round window with a brick frame. Doors, also topped with brick arches, are visible in the side fields. All these apertures are now covered with blind windows. A diagonal strip of windows along the eaves of the building was introduced to provide light to the rooms. The dimensions of the selected fragment of the building are 1740 × 123 × 1203 cm (Figure 2d).

All scans were performed during the summer months of 2021 on the campus of the Cracow University of Technology. During summer, trees sport full crowns of leaves and there is a lot of sun, thus creating demanding conditions for image acquisition and photogrammetric reconstruction. This is caused by lighting conditions and high contrasts and brightness, and by lush vegetation obscuring buildings. In order to properly compare the properties of the technologies under study, the scanning procedure with both the iPad device and photogrammetry was performed under the same lighting conditions. These conditions varied depending on the object (e.g., artificial light for the interior of the room, natural light for the garden furniture piece), and in the case of natural lighting, different acquisition conditions were tested—sunlight, shadow, or twilight.

### 2.3. Photogrammetric Reconstruction

Photogrammetric reconstruction entailed taking photographs by using a DSLM-type camera with a fixed focal length lens. Registration was carried out without the use of additional supporting markers. Photogrammetric reconstruction was performed by using Agisoft Metashape version 1.6.2 software [47], and each time the process went through multiple steps, starting with matching photographs, generating a sparse point cloud on their basis, creating a depth map for each of the matched photographs, and finally creating a dense point cloud.

Once the photographs were loaded into the program, their suitability for further processing was evaluated based on their sharpness. Algorithms used for feature extraction and descriptor creation resemble the approach known from the scale-invariant feature transform (SIFT) method [48]; however, due to its closed structure, the whole software should be treated as a black-box tool. The possibility of defining reconstruction parameters is limited to essential elements; however, the values of these parameters were constant for each performed reconstruction.

Agisoft Metashape usually generates very dense point clouds [49], which also contain redundant information in the form of noise. In order to eliminate it, the “confidence” parameter is used for each point cloud, which defines the number of occurrences of the given point in depth maps created from input images. The values of this coefficient depend on the number and coverage of input images; in practice, they usually do not exceed 10% of the number of input images. Therefore, points with high coefficient values are treated as more accurate and reliable, whereas those with low confidence levels can be rejected in the filtering process. It is assumed that points with confidence levels 1–2 are discarded; this significantly reduces the noise level in the photogrammetric reconstruction.

### 2.4. LiDAR Measurements with Apple iPad Pro

For this study, scans were obtained by using an Apple iPad Pro 12.9${}^{\prime \prime }$ 128 GB (Apple Inc., Cupertino, CA, USA) [43] running iOS version 15.0.1. Apple does not provide exact specifications of the sensors the device is fitted with. From the laconic statements available, it can be deduced that the sensor is based on ToF technology, and its range is about 5 m [43]. Through reverse engineering, independent researchers have determined that the LiDAR module consists of an emitter (vertical-cavity surface-emitting laser with diffraction optics element, VCSEL DOE) and a receptor (single-photon, avalanche diode, array-based, near infrared, complementary metal-oxide-semiconductor image sensor, SPAD NIR CMOS) based on direct-time-of-flight technology [44].

Registration quality with LiDAR sensors is only slightly affected by lighting conditions, so 3D scanning under varying natural light conditions in outdoor applications is possible while respecting possible imperfections in the quality of the resulting textures [45]. This is quite a promising property compared to photogrammetric techniques, where unfavourable lighting conditions, such as harsh shadows or backlighting, negatively affect the quality of the reconstructions obtained [50].

Of the vast and growing range of applications using the LiDAR sensor for 3D scanning, four were selected after initial testing: 3D Scanner App [51], Polycam [52], Scaniverse [53], and SiteScape [54]. After many tests, the Scaniverse app was selected for further research as it allows us to obtain 3D models in form of point clouds or meshes with high accuracy. Due to the use of a LiDAR sensor, it is supported by the following devices: the iPhone 12 Pro, the iPhone 12 Pro Max, the iPhone 13 Pro, the iPhone 13 Pro Max, and the iPad Pro (2020, 2021). Scaniverse allows visualising scanned models both in 3D and directly in augmented reality. The app offers a PRO version that exports high-resolution models to the following formats: FBX, OBJ, GLB, USDZ, STL, PLY, and LAS. Scaniverse provides an attractive option in the ability to set the scanning range to skip the area that is not needed. In the case of small models, the maximum scanning range of 5 m may be unnecessary because there is no need to scan such a large area but only a specific object. After scanning the model, Scaniverse gives the option to save it in one of three resolutions: standard (2k texture and 12-mm grid), high (4k texture and 8-mm grid), and ultra (8k texture and 6-mm grid). It is also possible to edit the scanned model within the application. We can improve its appearance or trim it. Another essential feature is the ability to measure the actual size of the scanned model. All these features contributed to the fact that Scaniverse proved to be the most adequate to achieve the research aim among many available applications.

### 2.5. Reference Data

The results were verified by using two approaches. For objects with highly complex geometries (sculptures at different scales), 3D scans were used. For geometricised, cuboid objects, 3D CAD models were developed.

A Konica–Minolta scanner was used to create a reference surface model of the gypsum figurine (Figure 3a). The boundary dimensions (described in Section 2.1), were also measured by using callipers. The heights of all three components of the sculpture were measured and found to have the following heights: 80.5, 90.0, and 84.4 mm. The measurement accuracy was 0.1 mm.

The second object, a sculpture on a pedestal, located in a park, also had a complex geometry. The Creaform Academia 3D structural slight scanner was used to procure its measurement data. The surface model produced had an accuracy of 1 mm. The reference mesh featured over 3.6 million vertices (Figure 3b).

CAD techniques worked very well for modelling the interior of the lobby room. A laser rangefinder and a measuring tape were used as measuring tools. The measurement accuracy was 1 cm (Figure 3c).

The same technique and tools were used to generate a model of part of the building. In this case, there was limited access to the upper parts of the facade. The surface model was compared to a point cloud in FLS format obtained from a laser scan. The scan, which was obtained courtesy of Bimtelligent (www.bimtelligent.pl, accessed on 4 October 2021), was taken with the scanner positioned approximately 7.3 m from the wall, in the middle of the wall width. By superimposing the model on the point cloud, the points of the surface model located in the higher parts of the building could be verified and considered reliable (Figure 3d).

### 2.6. Fit Measures Used

Comparisons between models were performed by using CloudCompare software. The distance from the model grid surface was determined for all photogrammetric reconstruction points and iPad LiDAR scans. This resulted in a root mean square (RMS) distance matching measure. This measure can be formalized by using the formula
(1)$$D_{RMS}=\frac{1}{n}\sum_{i=1}^{n}\sqrt{\left|p_{i}^{2}-m_{i}^{2}\right|}$$
where: *n*—the number of reconstruction points, $p_{i}$—reconstruction point, $m_{i}$—point closest to $p_{i}$ on the reference model.

The distance distribution $d_{RMS}$ for each reconstruction point is also used for analysis. Histograms showing the fit of the reconstruction to the model and positional statistics about the extent of the fit are also built from this.

### 2.7. Methods of Statistical Analysis

The statistical comparison method used was described at length by Labedz et al. [36]. In this method, it is assumed that the analysed variable is positional: the smaller the distance between the obtained point cloud and the reference model, the more correct the result. The statistical values that are considered are the quartiles and the interquartile range. Subsequent quartiles provide information about how far away from the original model the 25% ($Q_{1}$), 50% ($Q_{2}$, median), and 75% points ($Q_{3}$) are. These are calculated based on the absolute distances of the cloud points from the reference model, so the side they are on (in front of the model or behind the model) is not taken into account. Such information is of little importance in the context of mapping correctness. The smaller the value of the considered quantities, the better the result because more points lie closer to the reference model. Another analysed value is the interquartile range (IQR), which is a measure of dispersion. Unlike quartiles, it is calculated on distance data with the sign and is defined as the difference between the third and first quartiles: $IQR=Q_{3}-Q_{1}$. From this definition, it follows that 50% points lie at a distance from the model, defined by the IQR,value so that it can be treated as a measure of diversity; a narrower interval means less diversity in the variable being analysed. In the case presented, this means a greater concentration of points closer to the reference model, and thus a better score. The last statistical value examined is the number of points lying outside a particular distance range from the reference model. The average $Q_{3}$ values calculated on the unsigned data were taken as this distance (sigma), so these are points whose distance is significant from the reference data. A larger number of such points indicates a lower accuracy of the obtained experimental data [36].

The value can be easily determined directly, namely the number of points in the resulting cloud, which is also not without significance. Photogrammetric methods tend to generate very dense clouds, which may or may not indicate a higher model accuracy. Conversely, a small number of points for models of significant size may indicate a high degree of data generalisation.

## 3. Results

The object of this study was to compare 3D scans acquired by using the LiDAR sensor of the Apple iPad Pro with photogrammetric reconstructions. This was to verify the measurement capabilities of the device when applied to specific object groups. The experimental data collected by photogrammetric methods and with the device mentioned above were compared with the reference data presented in Section 2.5. The dense point cloud created by using photogrammetric methods and the cloud created by using the Apple iPad Pro were recorded by using the ICP method [55] to compare the datasets. The distance between the analysed cloud and the reference model was then calculated for each cloud. The resulting data was used to perform both visual and statistical analyses.

### 3.1. Measuring Accuracy of the iPad LiDAR Sensor

The first step in testing the iPad LiDAR sensor was to compare measurement accuracy by using reference spheres. For this purpose, we used reference spheres provided by the Laboratory of Coordinate Metrology of Cracow University of Technology (Figure 1). The calibration set consisted of two matte metal balls with calibrated diameters whose centres were located at the distance $d_{k}=269.4$ mm. Our goal was to estimate the distance between the spheres based on iPad LiDAR scans and compare the readings with the nominal value $d_{k}$. For this reason, 20 scans of the surface of the reference spheres were taken, of which two scans were rejected due to discontinuities in the reconstructed surface. This number is redundant, as seven measurements are sufficient to assess the measurement accuracy of a device [56]. Zeiss GOM software was then used to compare the sphere equations to the scanned surfaces. In each case, the distances between the centres of the matched spheres were determined (Figure 4). From these 18 measurements, the mean distance between the sphere centres $d_{avg}$ and the standard deviation std were estimated: $d_{avg}=270.81$ mm, std=4.06. As a result, the difference of the mean value $d_{avg}$ to the nominal value $d_{k}$ was determined: $\Delta_{k}=1.41$ mm, which was 0.52% of the nominal distance $d_{k}$.

### 3.2. Comparison to the Model

Model preparation based on experimental data sometimes involved repeating the acquisition process many times. Although the procedure of collecting photographs for photogrammetric reconstruction has been known and studied for years, the use of an Apple iPad Pro equipped with a LiDAR sensor required multiple attempts to acquire the necessary experience. The first necessary element of the procedure was the selection of the appropriate software, described in Section 2.4, with which the acquisition procedure was then performed. During its execution, many scanning attempts failed. This was particularly evident in the case of the fine detail and the small architectural object, for which up to a dozen attempts were necessary.

The data obtained were subjected to a preliminary visual verification. At this stage, models containing coarse errors in spatial data were rejected (Figure 5).

The original purpose of the iPad distance sensor predestines it for measuring medium-sized objects and interiors up to 5 m away from the sensor. For small objects of approximately a few centimetres, the measurement density is insufficient, and results in the occurrence of distortions and structure discontinuities (Figure 5a–b). For larger objects, which are difficult to cover with the sensor’s range, errors related to the acquisition mechanics occur, caused by quick or abrupt sensor movement during data capture, resulting in the appearance of unnatural shifts of the structure and duplication of fragments (Figure 5c–d).

In the next stage of verification, the models were evaluated in IQR and $Q_{3}$ values, and those with the best IQR and $Q_{3}$ values were selected for the final presentation. A similar procedure was carried out for data collected by using photogrammetric reconstruction, but in this case, no data contained coarse errors, and the number of acquisition repetitions was a maximum of 3.

#### 3.2.1. Figure Three Wise Monkeys—Fine Detail

The smallest model that was considered was a small gypsum figurine. It was characterised by a significant number of fine details and surface irregularities, which presented a challenge for data acquisition methods. The visualisations clearly show the difference in data density between the photogrammetric model and that coming from the Apple iPad Pro (Figure 6). The cloud derived from the iPad has apparent regularity features (Figure 6e), and the grid constructed from it is characterised by significant data generalisation, leading to a loss of a fair amount of detail. This is particularly evident in the right part of the surface model (Figure 6d). Interestingly, the distances of the measurement points obtained from the iPad device from the reference model were not very large. A total of 25% of them were less than 1 mm away from the model, and another 25% were more than 3 mm away (Table 1). By carefully analysing the distribution of the cloud points, it could be observed that the worst results were obtained at the locations of the various depressions of the model (cf. Figure 6d,e), whereas relatively good results were obtained at the locations of the convexities. It should be reiterated that the presented model was the best one obtained by using the Apple iPad Pro.

The photogrammetric reconstruction of the figure shown has surprisingly high accuracy (Figure 6a–c). It was obtained from only 19 photographs. The number of points that can be considered to lie at a considerable distance from the reference model is very negligible (Table 2). However, it should be noted that the σ value of 0.15 cm, in this case, was influenced according to the methodology adopted by both the high $Q_{3}$ value for the iPad model and the very low value for the photogrammetry. Not surprisingly, more than 50% of the points were outside the σ value for the iPad model. The data presented indicate that the Apple iPad Pro is unsuitable for imaging fine architectural details and ornaments.

#### 3.2.2. Garden Architectural Object-Bust on Plinth

The bust of T. Kosciuszko, placed on a pedestal, served as an example of a small built object. It is an object with a relatively complex surface, a non-convex structure and multiple grooves and concavities. This proved challenging to model. The Creaform Academia universal structured light handheld 3D scanner was used for this purpose. The object surface was reconstructed with an accuracy of 1 mm. The outdoor location of the sculpture, among trees, made it problematic to ensure reasonably uniform lighting conditions. The influence of light reflections was particularly troublesome when acquiring images for photogrammetric reconstruction. For the iPad LIDAR sensor, it was very problematic to scan the surface of the plinth, which was covered with a glossy granite pattern that introduced distortions in the cover reading.

Due to the varied colours of the surface details, the photogrammetric reconstruction (Figure 7a,b) was able to produce a high-resolution model (approximately 200,000 points). Unfortunately, the need to acquire the images in the evening due to better lighting conditions resulted in poorer colour reproduction (Figure 7a). Mapping the object by using the iPAD Pro LiDAR resulted in a much sparser, but also more regular point cloud structure (Figure 7e). The textured mesh showed surface discontinuities and irregularities (Figure 7d), which nevertheless did not deform it as drastically as observed for the small object (Figure 6d). The discontinuities and minor duplications seen in the eye and mouth area are related to the deformation of the texture, not the geometric mesh (Figure 7d). In this area, the mesh matched the reference model quite well, as illustrated by the blue points in Figure 7e.

The point cloud obtained with photogrammetric methods was much sparser than the ones obtained in the other analysed cases, yet its density was still more than 250 times higher than the one obtained with the iPad (Table 3). In the statistical data, we can observe similar accuracy of both methods for the first quartile and the median. For the iPad data, these are the points lying on the surface of the plinth walls. On the other hand, the data for the third quartile, as well as the sigma value for the iPad (Table 3) show the existence of a considerable number of inaccurately reconstructed points, which can be referred to as the already mentioned sharp edges. However, it is noteworthy that above the 3σ value of 2, i.e., above a 2-cm distance from the reference model, the number of these points is not so significant, and above 3, there is not a single point for the iPad data (Table 4).

#### 3.2.3. CUT Building Hallway–Interior

The indoor space, a fragment of a hall leading to a series of classrooms, had a slightly different character than the other analysed models. First of all, it was an enclosed space, which made it difficult to present its overall appearance on visualisation drawings. For this reason, it was decided to present its fragment as separated diagonally without taking into account the data for the ceiling. The size of the room (690 × 525 × 275 cm) and the multitude of difficult-to-map elements made it necessary to move along the walls, as well as to penetrate recesses and niches during data acquisition with the Apple iPad Pro. This type of measurement procedure may have caused a build-up of errors when performing multiple scans of certain sections of the room, which could not be avoided due to the room’s design. This resulted in the rejection of several obtained models due to coarse errors such as double-wall geometry. Another problem was the mixing of different types of lighting, i.e., artificial and natural, coming from behind glass doors.

On the presented visualisation (Figure 8b,d), one can again notice a significant difference in the density and regularity of the obtained point cloud. This is confirmed by the numerical data (Table 5), which show that the number of cloud points derived from the photogrammetric reconstruction is as much as 220 times greater. The mapping accuracy is already suggested by the shape of the histograms, whereas for the iPad data (Figure 8f), it is much more strongly stretched toward higher values than the histogram for the photogrammetric data (Figure 8c). Again, confirmation can be found in the statistics (Table 5): 75% of the cloud points obtained from photogrammetry lie within 2.8 cm of the model. For the cloud from the iPad, this value is 6.6 cm. It should be noted that compared to the room size, these values are not large—0.41% and 0.96% of the largest dimension, respectively. However, the data spread was almost three times larger for the cloud coming from the Apple iPad Pro.

The regularity of the point cloud obtained with Apple iPad Pro made the quality of the mesh based on this cloud much higher than the mesh based on data from photogrammetry (Figure 8a,c). The photogrammetric data showed various distortions on smooth surfaces and insufficiently illuminated areas. The iPad data acquisition procedure, which required recess penetration, resulted in a much more accurate reproduction of such elements.

Analysis of the deviations beyond the σ value (Table 6) of 5 cm, in this case, led to interesting conclusions. Although for the data collected with the Apple iPad Pro the number of points above the sigma value was significant (more than 41%), the number of points above the 2σ value decreases significantly, taking on acceptable values at 3σ. For photogrammetry, these values were shallow in all ranges.

#### 3.2.4. Building Facade

The building facade was the largest object selected for analysis. As with the small architectural object, data collection had to occur under outdoor conditions that had to be consistent for both acquisition methods. The first observation highlighted from the models obtained was the inability to scan the entire facade by using the Apple iPad Pro (Figure 9). This was due to the range of the LiDAR sensor, which is at most about 5 m. For this reason, in further consideration, we have limited the size range of the analysed object to the parts visible on all reconstructions (Figure 10). However, the quality of the acquired data was not affected by the atmospheric conditions during the acquisition. Both under harsh sunlight and overcast conditions, the device coped correctly with geometry mapping, although there were different textures. For photogrammetry, the soft light of an overcast day provided better acquisition conditions. In the visualisation shown (Figure 10b), it can be seen that the representation of cavities posed the biggest problem for both methods. Here, the distance from the reference model was the largest.

The histograms of the distribution of the distances of the collected experimental data from the reference model had a reasonably similar shape (Figure 10c,f). However, it should be kept in mind that the number of points in each model differed by at least two orders of magnitude (Table 7), which was observed in all the presented examples. The difference in the density of the obtained data is also clearly shown in Figure 10b,e.

When analysing the data from the statistical side, one can notice a very similar value of the $Q_{1}$ 1 coefficient for both experimental models (Table 7). This means that 25% of the reconstructed points were at a distance of about 1 cm from the reference model. Considering the dimensions of the model, these results should be considered good. Compared to the room model, the values of the statistical coefficients were stable and close to each other. The difference in the coefficient IQR was insignificant, with more than 300% in the earlier example. Moreover, the analysis of the number of outlying points (Table 8) showed that the accuracy of data representation for both methods was similar and relatively high. For the σ value (which is also 5 cm in this case), the number of points above it for both the photogrammetric and iPad reconstruction were at a similar level of about 20%, so in this case, the differences can be considered marginal.

The facade of the building that was chosen for the experiments featured moderate geometric complexity. Some of the architectural details were fairly fine; nonetheless, the cavities on the elevation were not too deep. Moreover, there were no nooks and recesses that could not be accessed by the measurement devices. It should be noted that there is a great variety of architectural styles, ranging from those characterised by the simplest form (such as modernism) to the most formally complex (such as baroque). For the latter, the approach suited to small-scale details and garden sculpture would be more appropriate, with the caveat, however, that the rear parts of sculptures adjacent to the facade may be impossible to represent in models.

## 4. Discussion and Conclusions

Models equipped with various types of ToF sensors are starting, increasingly prominently, to enter the market of mobile consumer devices. Their role is to support the acquisition of depth maps and thus improve the obtainment of digital 3D models of the surrounding reality. They can be used in a variety of ways. One such device is the Apple iPad Pro, which comes equipped with a LiDAR sensor based on dToF technology. The presented research aimed to verify the measurement capabilities of this device concerning both reference models and models derived from photogrammetric reconstruction or laser scanners. Four types of architectural objects were selected to perform this research.

The first notable element when preparing spatial data acquisition using the mentioned device is the multitude of applications that offer such functionality. It was not the authors’ intention to compare different applications, so one of them was chosen after preliminary tests: Scaniverse. The process of data acquisition was carried out without the control of lighting conditions, i.e., they were not selected for a specific application. Obtaining the optimal data acquisition result by using the Apple iPad Pro was not an easy task. Sometimes, several attempts were necessary for the model to meet expectations, at least to some extent. The most frequent errors concerned geometry multiplication and the incorrect merging of adjacent model fragments. Additionally, the accumulation of measurement errors made during the scanning process was a characteristic effect.

Depending on the applications and methods used, the result of spatial data acquisition can be both point clouds and polygonal meshes, the topology of which is derived from the cloud structure. Both technologies produce grids with different densities and structures. Meshes derived from photogrammetry are dense, and their surfaces are rough. The meshes generated by an iPad are smooth and highly generalised. This observation was common to all four types of architectural objects tested.

Based on a statistically significant series of measurements made on reference spheres, the average measurement error for iPad LiDAR was tentatively estimated. Afterward, models obtained with it were compared with reference models to gauge accuracy. Statistical analysis showed that high cloud density derived from photogrammetry was also accompanied by high accuracy. The situation was slightly different for the data coming from the Apple iPad Pro. For most miniature objects (fine detail, ornament), the deviation analysis showed a high inconsistency with the reference models. For larger objects (interior, facade of a building), the statistics approached the values of the data derived from photogrammetry. However, it should be emphasised that the statistics were related to the distance from the reference model in the nodes of the regular grid derived from the experimental data. The already mentioned low data density and related generalisation negatively influence the quality of resulting data, especially in the case of small-sized objects and a high level of detail.

These properties imply the purpose of the technologies that were studied. Scanning with an iPad may be sufficient for data acquisition of objects with smooth surfaces or for which the rough surface structure is not essential. The generalisation it provides may be desirable in many situations. For example, in the case of a wall, the edge dimensions are crucial, not the surface texture, for an imperfection in a wall cavity, a floor, or ceiling surface. Therefore, the iPad works quite well for indoor space mapping. Unlike photogrammetry, the data from such a scan is already correctly scaled, and no loss of representation continuity is observed for colour-uniform surfaces. However, one should keep in mind the limitation in the range of the iPad’s LiDAR sensor, which is 5 m. This property significantly limits the usefulness of the device in field spatial data acquisition.

## Figures and Tables

**Figure 1 sensors-22-08504-f001:**
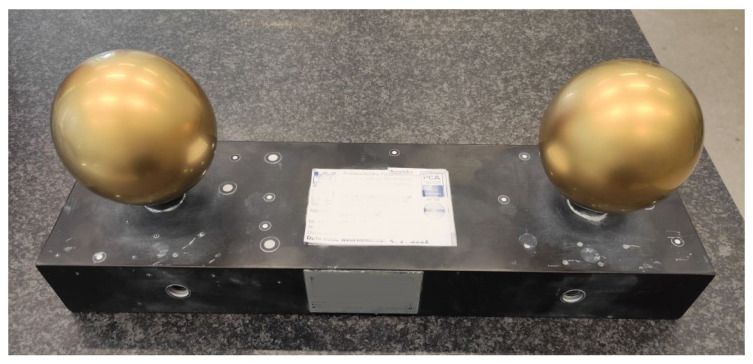
Calibration balls in the Laboratory of Coordinate Metrology, Cracow University of Technology.

**Figure 2 sensors-22-08504-f002:**
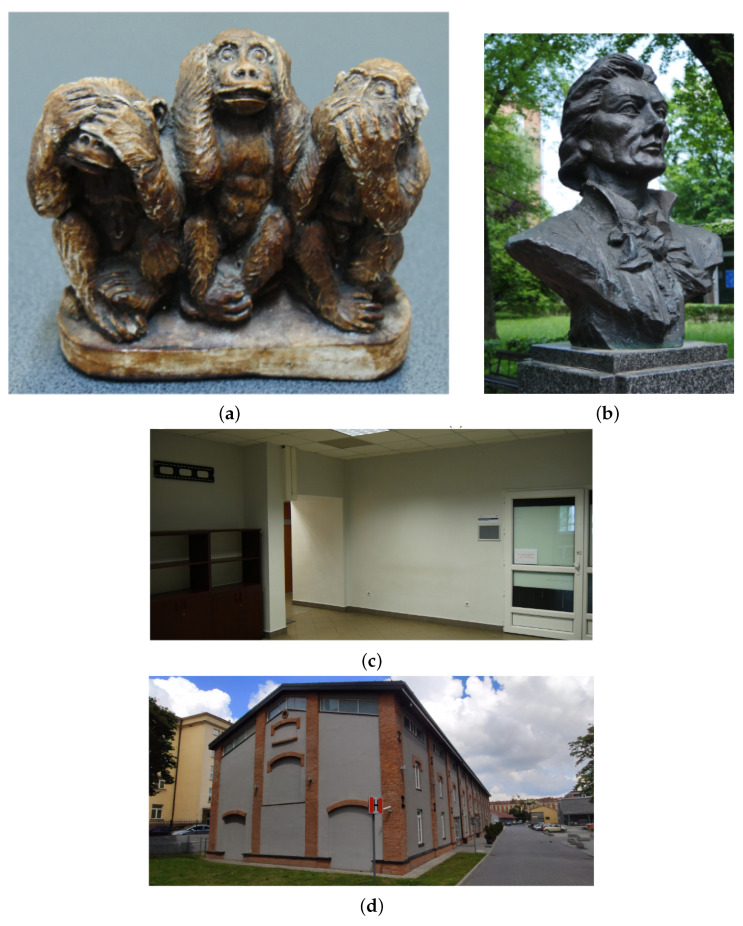
Objects at various scales used during the study: (**a**) fine detail; (**b**) small architecture object; (**c**) room interior; (**d**) building facade.

**Figure 3 sensors-22-08504-f003:**
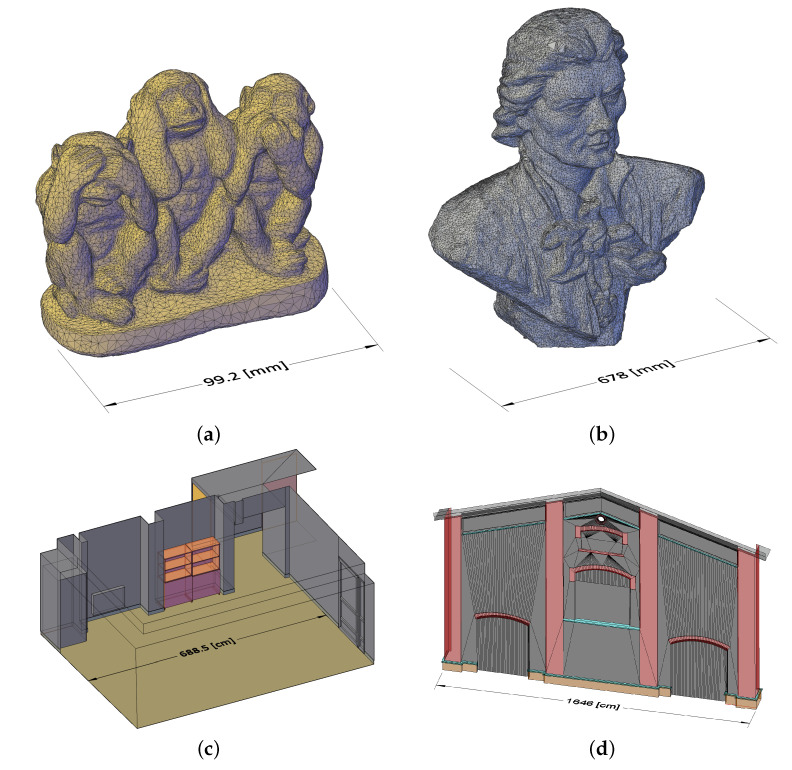
Reference models: (**a**) small figure model—*Three Wise Monkeys*, Konica-Minolta laser scanner; (**b**) medium model—bust of Tadeusz Kościuszko, Creaform Academia 3D scanner; (**c**) interior model—student lounge, CAD model; (**d**) facade model—CUT cannon shed building, CAD model.

**Figure 4 sensors-22-08504-f004:**
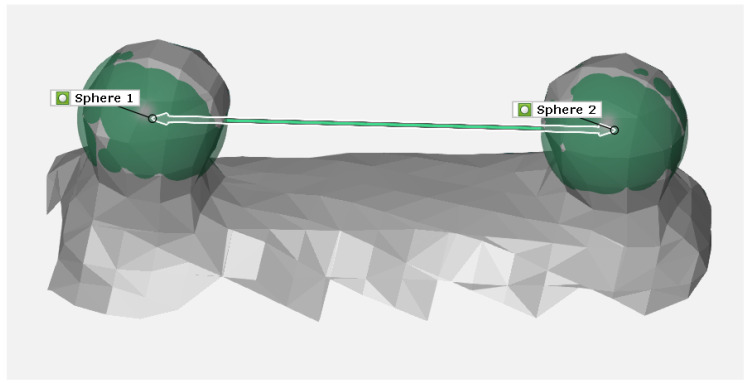
Measurements with calibration balls using GOM Software.

**Figure 5 sensors-22-08504-f005:**
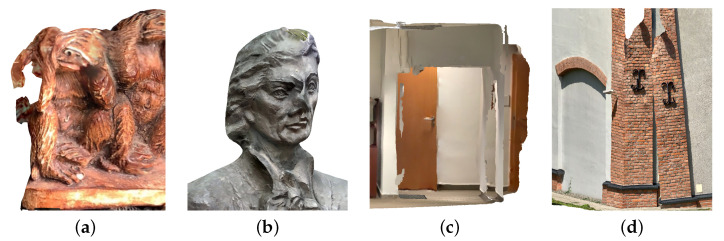
Common bugs in iPad meshes: (**a**)—discontinuity and deformation, (**b**)—deformation and displacement, (**c**)—duplication, (**d**)— duplication.

**Figure 6 sensors-22-08504-f006:**
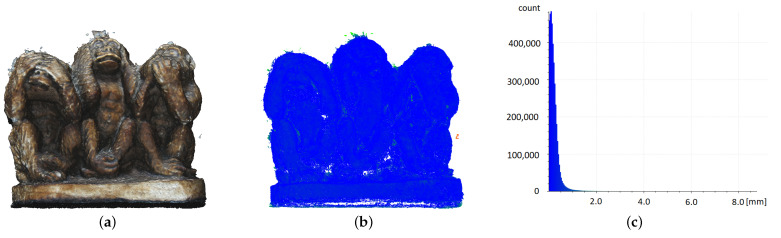
Comparison of statistical values for fine detail. Photogrammetry: (**a**) textured mesh, (**b**) point cloud, (**c**) histogram of distances to the model; iPad: (**d**) mesh, (**e**) point cloud, (**f**) histogram of distances to the model. Point clouds are coloured according to histogram data.

**Figure 7 sensors-22-08504-f007:**
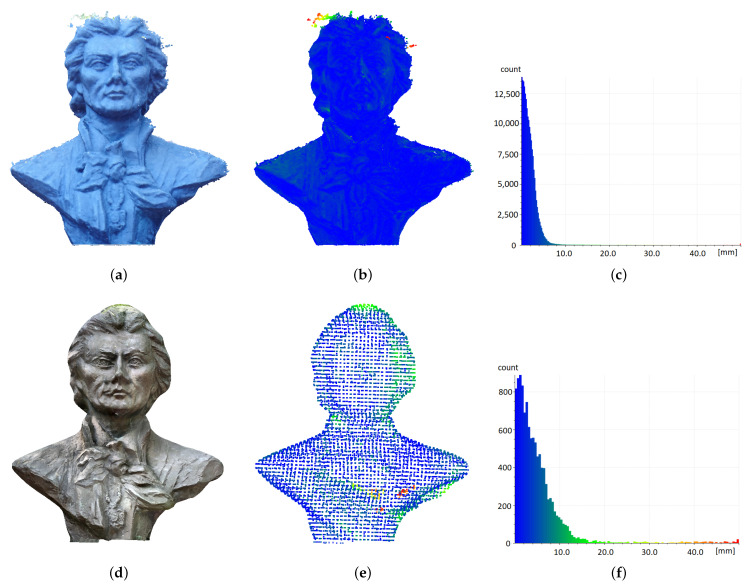
Comparison of measured data with the model for the small architectural object. Photogrammetry: (**a**) textured mesh, (**b**) point cloud, (**c**) histogram of distances to the model; iPad: (**d**) mesh, (**e**) point cloud, (**f**) histogram of distances to the model. Point clouds are coloured according to histogram data.

**Figure 8 sensors-22-08504-f008:**
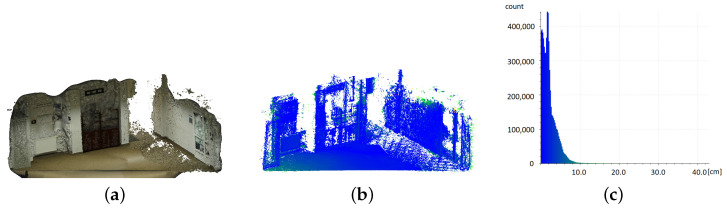
Comparison of measured data with model for room interior. Photogrammetry: (**a**) textured mesh, (**b**) point cloud, (**c**) histogram of distances to the model; iPad: (**d**) mesh, (**e**) point cloud, (**f**) histogram of distances to the model. Point clouds are coloured according to histogram data.

**Figure 9 sensors-22-08504-f009:**
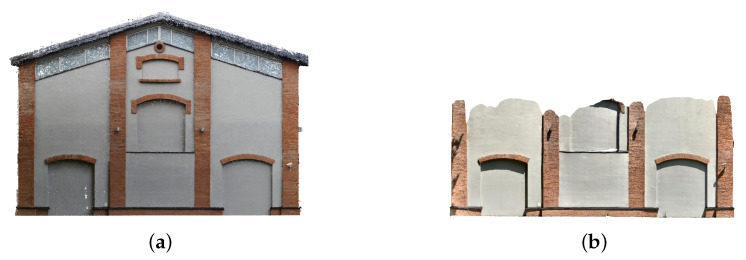
Measurement limits: (**a**) photogrammetry, (**b**) iPad LiDAR.

**Figure 10 sensors-22-08504-f010:**
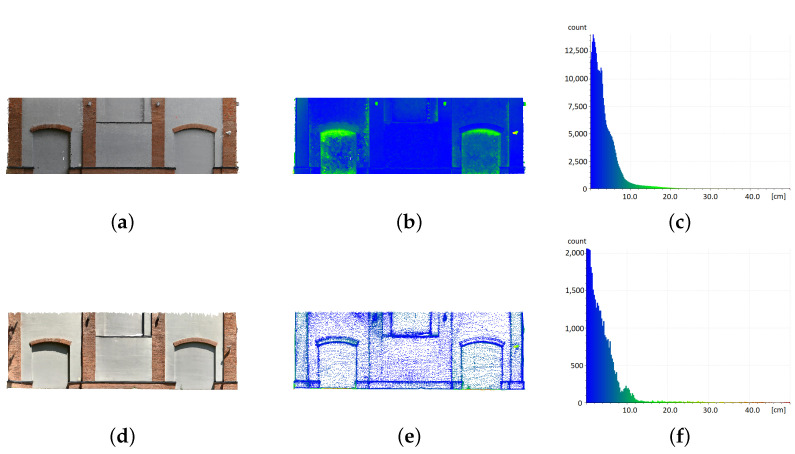
Comparison of statistical values for the building facade. Photogrammetry: (**a**) textured mesh, (**b**) point cloud, (**c**) histogram of distances to the model; iPad: (**d**) mesh, (**e**) point cloud, (**f**) histogram of distances to the model. Point clouds are coloured according to histogram data.

**Table 1 sensors-22-08504-t001:** Comparison of statistical values for fine detail [mm].

Model	Points	$Q_{1}$	Median	$Q_{3}$	*IQR*
iPad	1254	0.75	1.68	2.95	2.52
photogrammetry	3,622,007	0.08	0.16	0.27	0.25

**Table 2 sensors-22-08504-t002:** Analysis of the number of outlying points (σ = $Q_{3}$ = 1.5 [mm]) for fine detail.

Model	σ	[%]	2σ	[%]	3σ	[%]
iPad	665	53.03	293	23.37	116	9.25
photogrammetry	4874	0.13	589	0.02	296	0.01

**Table 3 sensors-22-08504-t003:** Comparison of statistical values for the small architecture object [mm].

Model	Points	$Q_{1}$	Median	$Q_{3}$	*IQR*
iPad	9625	1.56	3.47	6.13	7.0
photogrammetry	193,127	0.70	1.50	2.52	3.0

**Table 4 sensors-22-08504-t004:** Analysis of the number of outlying points (σ = $Q_{3}$ = 5 [mm]) for the small architectural object.

Model	σ	[%]	2σ	[%]	3σ	[%]
iPad	1630	16.94	449	4.66	76	0.69
photogrammetry	1799	0.93	620	0.32	460	0.24

**Table 5 sensors-22-08504-t005:** Comparison of statistical values for the room model [cm].

Model	Points	$Q_{1}$	Median	$Q_{3}$	*IQR*
iPad	27,776	2.0	4.2	6.6	7.7
photogrammetry	6,138,644	0.8	1.7	2.8	2.5

**Table 6 sensors-22-08504-t006:** Analysis of the number of outlying points (σ = $Q_{3}$ =5 [cm]) for the indoor space model.

Model	σ	[%]	2σ	[%]	3σ	[%]
iPad	11,401	41.05	1710	6.16	362	1.30
photo-grammetry	426,017	6.94	26,974	0.44	6830	0.11

**Table 7 sensors-22-08504-t007:** Comparison of statistical values for the building facade [cm].

Model	Points	$Q_{1}$	Median	$Q_{3}$	*IQR*
iPad	33,363	1.2	2.9	5.2	5.3
photogrammetry	3,031,591	1.2	2.5	4.5	5.1

**Table 8 sensors-22-08504-t008:** Analysis of the number of outlying points (σ = $Q_{3}$ = 5 [cm]) for the building facade.

Model	σ	[%]	2σ	[%]	3σ	[%]
iPad	6699	20.08	1905	5.71	806	2.42
photo-grammetry	574,692	18.96	104,422	3.44	39,174	1.29

## Data Availability

Upon a reasonable request from the corresponding author.

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
