# Peer review of "Accuracy Verification of Surface Models of Architectural Objects from the iPad LiDAR in the Context of Photogrammetry Methods"

_sensors, 2022, doi:10.3390/s22218504_

Round 1

Author Response

Dear Reviewer, 

Thank you very much for your time devoted to our work and for your positive review. All corrections in the text are marked in yellow.  

Kind regards 

Authors 

Reviewer 2 Report

none

Author Response

(The authors gave the same response as above.)

Reviewer 3 Report

The purpose of this paper is to compare  3D scans acquired using the AppleiPad Pro LIDAR sensor with photogrammetric reconstructions. Measurements accuracy was compared and four types of architectural elements are involved: 1)a small detail of significant geometric complexity, 2) a piece of garden furniture, 3) an architectural interior and 4) a building’s facade.

The method is well identified in the paragraph 2  with the Photogrammetric Reconstruction, the LIDAR measurements with Apple iPad Pro, Reference data, Fit measured data and method of Statistical analysis Why the fisrt figure is the 3° and not the 1a? The results are well defined.

Author Response

Dear Reviewer, 

Thank you very much for your time devoted to our work. We appreciate your very constructive comments. Thank you very much for your constructive comments. Figure 3a has been moved to the appropriate text position and numbered 1. All corrections in the text are marked in yellow.  

Kind regards 

Authors 

Reviewer 4 Report

Dear Authors,

Please find the comments, suggestions and doubts that arose.

1.      Pp. 2. Line 34, the abbreviation LiDAR is defined only up to line 48

2.      Pp. 2. Line 82. Figure 3a is called before the previous ones. Authors are encouraged to move this figure as it is the one mentioned first.

3.      Pp. 4. Line 155. The Authors claim that all scans were performed during the summer of 2021”.  Is there any special dependency on environmental conditions in the other months?

4.      Some references appear as links throughout the manuscript, please give the corresponding Ref.  number. For example, pp. 8 line 253.

5.      Figure 4 is ambiguously discussed. What are the origins of these bugs and rejection criteria beyond the relative visual selection?

6.      Figure 6d shows a certain deformation and displacement similar to what is observed in Figure 4b, which the Authors mention was rejected for containing coarse errors. Can you clarify this doubt? Some idea is given on pp. 11, lines 363-367 but the discussion is confusing.

7.      Interesting results are observed for Building Facade. But what happens if the facade has greater detail, to illustrate the idea, the facade of Saints Peter and Paul Church or related ones?

Author Response

Dear Reviewer, 

Thank you very much for your time devoted to our work. We appreciate your very constructive comments. All corrections in the text are marked in yellow. We provide answers to individual comments below. 

  1. Pp. 2. Line 34, the abbreviation LiDAR is defined only up to line 48 

We added the explanation in the line 34. 

  1. Pp. 2. Line 82. Figure 3a is called before the previous ones. Authors are encouraged to move this figure as it is the one mentioned first. 

We moved and renumbered this figure 

  1. Pp. 4. Line 155. The Authors claim that all scans were performed during the summer of 2021”.  Is there any special dependency on environmental conditions in the other months? 

We added the explanation: 

The season when the trees are leafy and the sun often operates very hard creates the most demanding conditions for image acquisition,  threfore, for photogrammetric reconstruction. On the one hand, this is due to the lighting conditions and high brightness contrasts, and on the other hand - the objects of analysis are partly obscured by vegetation. 

  1. Some references appear as links throughout the manuscript, please give the corresponding Ref.  number. For example, pp. 8 line 253. 

Some references are at the end of the sentence ora the paragraph. We added a numer in the line 253. 

  1. Figure 4 is ambiguously discussed. What are the origins of these bugs and rejection criteria beyond the relative visual selection? 

We supplemented the explanation of the sources of acquisition errors: 

The original purpose of the iPad distance sensor predestines it for measuring medium-sized objects and interiors up to 5 m away from the sensor. For small entities of approximately a few cm, the measurement density is insufficient, which manifests itself in the occurrence of distortions and structure discontinuities (fig. 5a-b). For larger objects, which are difficult to encompass with the span of the sensor, errors related to the mechanics of acquisition occur, caused by rapid or abrupt movement of the sensor during data capture, resulting in the appearance of unnatural shifts of the structure and duplication of fragments (fig.5 c-d). 

  1. Figure 6d shows a certain deformation and displacement similar to what is observed in Figure 4b, which the Authors mention was rejected for containing coarse errors. Can you clarify this doubt? Some idea is given on pp. 11, lines 363-367 but the discussion is confusing. 

We justified the choice more precisely: 

The discontinuities and minor duplications seen in the eye and mouth area are related to the deformation of the texture, not the geometric mesh (fig 7d). In this area, the mesh matches the reference model quite well, as illustrated by the blue points in Fig. 7e. 

  1. Interesting results are observed for Building Facade. But what happens if the facade has greater detail, to illustrate the idea, the facade of Saints Peter and Paul Church or related ones? 

We tried to refer to the other architectural examples:  

The facade of the building that was chosen for the experiments characterises by moderate geometric complexity. Some of the architectural details are fairly fine; nonetheless, the cavities on the elevation are not too deep. Moreover, there are no nooks and recesses that are inaccessible to the measuring devices. It is worth noting here, however, that there is a great variety of architectural styles, ranging from those characterised by the most simple form (such as modernism) to the most formally complex (such as baroque). For the latter, the approach described for the cases of small-scale detailing and garden sculpture would be more appropriate, with the caveat, however, that the rear parts of sculptures adjacent to the facade may be impossible to represent in models. 

Kind regards 

Authors 

Reviewer 5 Report

1. To more clearly point out the research gaps of existing studies.

2. To more clearly present the theoretical contributions of the works.

Author Response

Dear Reviewer, 

Thank you very much for your time devoted to our work. We appreciate your very constructive comments. We have tried to address them in Chapter 1 by expanding the description of the research gap and the contribution of our research. We have also completed the explanation of the research procedure in Chapter 4. We would also like to clarify that the article we have prepared is primarily practical rather than theoretical. All corrections in the text are marked in yellow.  

Kind regards 

Authors